# Semantic Impact–Driven Visual Scheduling for Vision Language Models

Xuan Wang [* 1]   Yilin Liu [* 1]   Fangxiang Feng [1]   Caixia Yuan [1]   Huixing Jiang [2]   Xiaojie Wang [1]

## Abstract

Vision-Language Models (VLMs) suffer from high inference latency due to long visual sequences. To enable efficient, on-demand utilization of visual information, we argue that visual necessity should be assessed by its semantic impact on the output distribution, rather than inferred from intermediate interaction signals such as attention weights. We propose a training-free framework based on token embedding subspace decomposition, which we term a prediction-conditioned *Semantic Lens*. Specifically, at fixed decoding intervals, we perform QR decomposition on the Top-K candidate token embeddings to construct an orthogonal semantic basis. We then introduce **S**emantic **I**mpact–Driven **V**isual **S**cheduling (SIVS), which measures how visual inputs impact model predictions by projecting visual-induced hidden-state variations onto this semantic lens. SIVS provides a geometrically grounded, impact-driven criterion for dynamic visual KV scheduling. Empirical results demonstrate that SIVS achieves 87% visual KV compression while maintaining over 99% of model performance.

## 1. Introduction

Building on the rapid progress of large language models (LMMs) (OpenAI et al., 2024; AI@Meta, 2024; Yang et al., 2024; 2025a), researchers have successfully extended their capabilities to the multimodal setting, giving rise to Vision Language Models (VLMs) as a central paradigm for vision–language understanding (Chen et al.; Liu et al., b; Li et al., a; Liu et al., a; Bai et al., a). A major source of this challenge lies in the rapid growth of visual context length: modern visual encoders often translate a single image into hundreds or even thousands of visual tokens (Wang et al.; Bai et al., b; Lin et al.), all of which are provided as conditioning context during autoregressive decoding. As decoding proceeds, the model repeatedly attends to this large visual context, leading to substantial memory usage and latency overhead. Consequently, inference efficiency in VLMs is increasingly constrained by GPU memory capacity and memory bandwidth rather than raw compute throughput.

To address this issue, a growing body of work has explored methods for reducing or adapting the effective visual context during inference. Existing approaches include visual token pruning (Chen et al., 2024), token merging (Zhang et al., 2025b), and visual KV management (Wan et al.). While these methods differ in their mechanisms, they share a common goal: limiting the amount of visual information that the model actively conditions on at each decoding step. In practice, this is often realized by modifying the visual Key–Value (KV) caches used in attention. However, most existing methods rely on local or heuristic importance signals, such as attention weights, feature similarity, or entropy-based uncertainty measures, to decide which visual tokens to retain.

Although effective in reducing the inference cost of VLMs, such heuristics (Xing et al., 2025; Yang et al., 2025b) provide only an indirect approximation of semantic relevance. In particular, they implicitly assume that local salience correlates with a token's influence on the model's semantic decision. This assumption is sometimes violated in tasks requiring fine-grained semantic grounding—such as optical character recognition (Liu et al., 2024; Singh et al.) or document question answering (Mathew et al.)—where visually subtle or low-attention tokens may exert a disproportionate effect on the predicted output. Conversely, visually prominent regions may attract strong attention while contributing little to the model's final decision.

A key limitation of existing approaches is the absence of an explicit semantic criterion for determining when visual information should remain part of the model's conditioning context. In an autoregressive transformer, the hidden state at each decoding step encodes a mixture of information: components that are critical for determining the next-token distribution, and auxiliary variations that are largely irrelevant to the prediction (Geva et al., 2021; Hewitt & Manning,

*Equal contribution  [1]School of Artificial Intelligence, Beijing University of Posts and Telecommunications, Beijing, China [2]Li Auto, Beijing, China. Correspondence to: Xiaojie Wang <xjwang@bupt.edu.cn>.

*Proceedings of the 43$^{rd}$ International Conference on Machine Learning*, Seoul, South Korea. PMLR 306, 2026. Copyright 2026 by the author(s).

2019). Visual inputs can perturb the hidden state along both directions. However, only perturbations aligned with the model's semantic decision structure influence token ranking, while orthogonal variations—despite potentially large magnitude—have negligible effects on the output distribution. Methods that do not explicitly distinguish between these components risk either discarding semantically critical visual information or retaining visually salient but semantically irrelevant context. This calls for a mechanism that can "read out" hidden-state perturbations through the lens of the model's instantaneous semantic decision structure, rather than raw activation magnitude.

In this work, we propose a semantic subspace–guided framework for visual context selection, built upon a prediction-conditioned measurement mechanism that we term a *Semantic Lens*. Our approach is based on the observation that the model's instantaneous semantic intent can be approximated by the subspace spanned by the embeddings of its Top-$K$ predicted tokens. Given the predicted token embedding matrix $M_K \in \mathbb{R}^{K \times d}$, we obtain an orthogonal semantic basis via QR decomposition and use it to characterize decision-relevant semantic directions at each decoding step. We quantify the effect of visual conditioning by computing the hidden-state difference induced by visual inputs and projecting this difference onto this subspace. The hidden state is taken from a fixed intermediate layer, as prior work has shown that cross-modal information transfer and alignment in VLMs emerge predominantly within specific layers or layer ranges, rather than uniformly across the network (Zhang et al., 2025c; a). The projection measures the semantic impact attributable to visual information, separating it from non-semantic variations.

With quantified semantic impact, we decompose the semantic impact into token-level contributions by projecting each visual token's attention-induced update onto the lens-defined semantic correction direction. This enables fine-grained and interpretable selection of visual tokens, rather than relying on coarse, global heuristics. Our method maintains visual conditioning throughout decoding, while periodically reassessing its semantic contribution at a fixed interval of $T$ steps. This semi-dynamic design enables efficient reuse of previously computed semantic evaluations as decoding progresses, while preserving the consistency of recently acquired visually grounded semantics and avoiding disruptions to contextual coherence during decoding.

In summary, this work makes the following contributions.

- We introduce a Semantic Lens–based perspective on visual context selection, reframing visual token retention as a function of its influence on the model's instantaneous semantic decision rather than intermediate activation heuristics.

- We propose a training-free, inference-time framework for Semantic Impact–Driven Visual Scheduling (SIVS), which quantifies the decision-relevant necessity of visual tokens using a prediction-aligned semantic subspace. This formulation enables token-level attribution of semantic contributions, supporting fine-grained and interpretable visual context control during autoregressive decoding.

- Extensive experiments show that our method substantially reduces effective visual context length and associated KV cache usage, while preserving—and in some cases improving—downstream task performance, highlighting semantic alignment as a key principle for efficient multimodal inference.

## 2. Related Work

The rapid growth of visual context length in multimodal large language models has raised significant concerns regarding memory consumption and decoding latency. As modern models encode images into hundreds or even thousands of visual tokens, a growing body of work has explored how to reduce, adapt, or restructure the effective visual context during inference. Existing approaches can be broadly categorized into visual token pruning, token merging, and dynamic allocation of visual representations.

**Visual Token Pruning.** Visual token pruning exploits the spatial redundancy of visual representations by discarding tokens with low estimated importance. Early work, such as FastV (Chen et al., 2024), observes that attention over visual tokens becomes highly concentrated in deeper layers, enabling training-free pruning based on accumulated attention scores. Subsequent methods improve robustness through automated hyperparameter selection or progressive pruning (Xing et al., 2025). More recent approaches incorporate textual context into pruning decisions by conditioning token importance on both visual features and user prompts, or perform early-stage token reduction directly in the vision encoder before features are passed to the LLM (Zhang et al., 2025b;a). However, these approaches rely on attention-, similarity-, or encoder-side heuristics and typically perform irreversible token removal, which can discard information that becomes semantically important as the model's focus shifts during decoding.

**Token Merging and Aggregation.** Instead of discarding tokens, token merging methods reduce sequence length by aggregating similar visual tokens. VisionZip (Yang et al., 2025b) distinguishes dominant tokens encoding fine-grained object information from contextual tokens representing background structure, merging the latter into compact representations. Other approaches further extend similarity-

driven token fusion to multimodal settings (Bolya et al., 2023; Tan et al.). However, merging decisions are still typically driven by visual similarity or heuristic importance measures, and once tokens are fused, their individual semantic contributions become entangled, limiting the model's ability to adapt to evolving semantic demands during decoding.

**KV Management.** Beyond input-level pruning, KV cache eviction aims to bound the memory usage during autoregressive decoding. In the text domain, methods like H2O (Zhang et al., b) and SnapKV (Li et al., b) exploit attention sparsity to retain only salient tokens or clusters within a fixed-size cache. Extending these ideas to multimodal settings is more challenging due to the dense and spatially redundant nature of visual features. LOOK-M (Wan et al.) adopts a text-prior eviction strategy to selectively reduce the visual KV cache while preserving visually important tokens for multimodal interaction. Nevertheless, these eviction strategies are still guided by attention statistics or structural roles rather than a token's direct contribution to the model's semantic prediction, and removed KV entries cannot be recovered if they later become semantically relevant during decoding.

## 3. Method

We propose Semantic Impact–Driven Visual Scheduling (SIVS), a training-free framework for semi-dynamic visual KV scheduling during autoregressive decoding.

Unlike prior methods that rely on attention magnitudes or static sparsification heuristics, SIVS explicitly quantifies, through a Semantic Lens, how much visual information alters the model's semantic decision space and attributes this effect to individual visual tokens.

### 3.1. Overview and Inference Protocol

---

**Algorithm 1** Semantic Impact–Driven Visual Scheduling

**Require:** Visual tokens $\boldsymbol{V}$, update interval $T$, layer $\ell^\star$, Top-$K$ size $K$, threshold coefficient $\alpha$
1: Initialize $\Phi^\star \leftarrow \{1, \ldots, N\}$
2: **for** decoding step $t = 1, 2, \ldots$ **do**
3:  **if** $(t - 1) \bmod T = 0$ **then**
4:   Compute $\boldsymbol{h}_t^{(v)}$ and $\boldsymbol{h}_t^{(\emptyset)}$ at layer $\ell^\star$
5:   Extract Top-$K$ candidate embeddings $\boldsymbol{M}_K$
6:   Compute semantic basis $\boldsymbol{Q}$ via QR decomposition
7:   Compute $\boldsymbol{\delta}_{\text{sem}} = \boldsymbol{\Pi}_{\text{sem}}(\boldsymbol{h}_t^{(v)} - \boldsymbol{h}_t^{(\emptyset)})$
8:   Compute token scores $\{s_i\}$ and threshold $\tau_{\text{kv}}$
9:   Update $\Phi^\star = \{i \mid s_i > \tau_{\text{kv}}\}$
10:  **end if**
11:  Decode next token using visual KV indexed by $\Phi^\star$
12: **end for**

---

We consider an autoregressive multimodal Transformer with $L$ layers. At decoding step $t$, let $\boldsymbol{V} = [\boldsymbol{v}_1, \ldots, \boldsymbol{v}_N]$ denote the visual tokens. SIVS operates on an intermediate layer $\ell^\star$, which empirically balances multimodal fusion and semantic abstraction. We denote the hidden state at this layer as

$$\boldsymbol{h}_t^{(\ell^\star)} = f_{\ell^\star}(\mathbf{KV}_{\text{text}}, \mathbf{KV}_{\text{vis}}, \boldsymbol{q}_t). \tag{1}$$

SIVS operates entirely at inference time and introduces no parameter updates. The overall procedure is summarized in Algorithm 1. As illustrated in Figure 1, the method is executed in a *semi-dynamic* fashion: visual token selection is refreshed once every $T$ decoding steps and reused in between. At each update step $t$, SIVS performs the following stages:

1. **Semantic Impact Quantification**: measure the semantic impact $\boldsymbol{\delta}_{\text{sem}}$ induced by visual information through a semantic lens projection.

2. **Token-Level Attribution**: decompose $\boldsymbol{\delta}_{\text{sem}}$ into contributions of individual visual value tokens.

3. **Adaptive Selection**: retain only visually semantic tokens and reuse them for the next $T$ steps.

This design exploits the observation that semantic dependence on visual information evolves slowly across decoding steps, yielding substantial computational savings without sacrificing semantic fidelity.

### 3.2. Part I: Quantifying Semantic Impact

A naive approach to measuring visual influence is to contrast vision-aware and vision-blind hidden states at a pivotal intermediate layer $\ell^*$ (where $1 \ll \ell^* \ll L$). Specifically, we define the raw difference at layer $\ell^*$ as $\boldsymbol{h}_t^{(v)} - \boldsymbol{h}_t^{(\emptyset)}$ where the two states are computed as:

$$\boldsymbol{h}_t^{(v)} = f_{\ell^\star}(\ldots, \text{KV}_{\text{vis}}), \quad \boldsymbol{h}_t^{(\emptyset)} = f_{\ell^\star}(\ldots, \emptyset), \tag{2}$$

corresponding to vision-aware and vision-blind decoding, respectively. The choice of $\ell^*$ follows the principle that multimodal alignment is most pronounced in a stable mid-to-late layer range; we provide a detailed discussion of this selection criterion and supporting empirical analysis in Sec. 4 and the Appendix B.2. This layer serves as the primary locus where multimodal alignment and cross-modal information exchange predominantly take place (Zhang et al., a; 2025c). However, this raw difference still contains substantial non-semantic variations that do not affect the model's prediction. To isolate semantically effective changes, we must account for the model's instantaneous decision space.

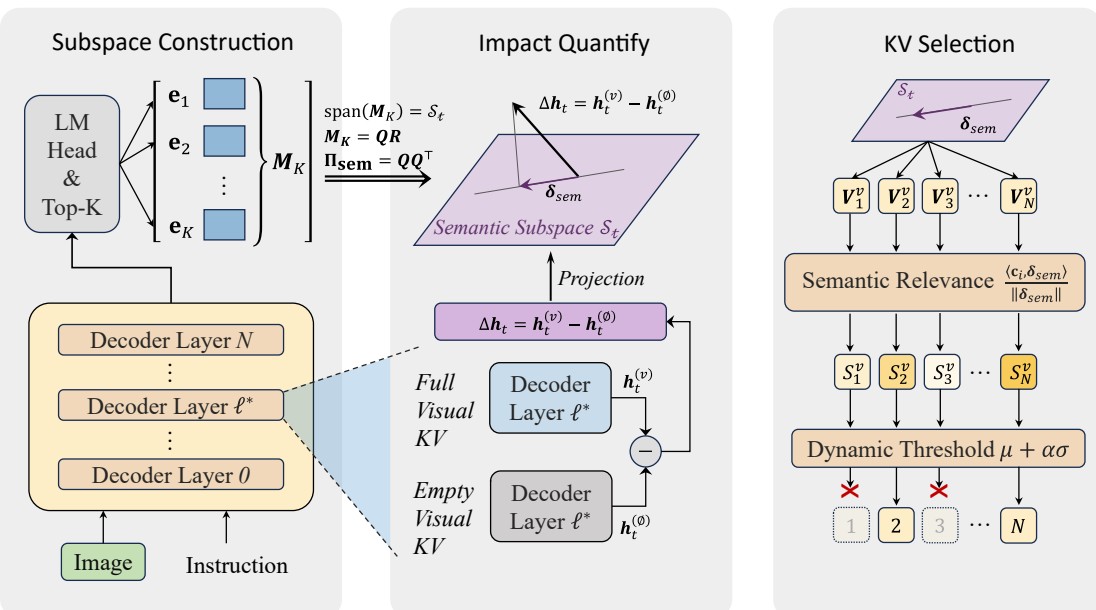

*Figure 1.* At selected decoding steps, SIVS contrasts vision-aware and vision-blind hidden states at an intermediate layer to obtain a semantic impact. This impact is projected onto a prediction-conditioned semantic subspace constructed from Top-$K$ candidate token embeddings, which serves as a *Semantic Lens* for measuring decision-relevant visual impact. The resulting semantic impact is then decomposed into token-wise contributions, enabling adaptive visual KV selection.

**Prediction-conditioned semantic subspace.** Although the model defines a distribution over the entire vocabulary, an autoregressive decoding step does *not* effectively "decide" among all tokens. In practice, the next-token outcome is determined by a small subset of candidates that dominate the probability mass and compete in logit space. Moreover, prior analyzes such as the logit lens have shown that hidden representations in the middle-to-late layers of large language models already encode coarse but predictive semantic information, enabling approximate semantic decomposition in the middle-late layer $\ell^*$ before the final layer (Hewitt & Manning, 2019; nostalgebraist, 2020; Belrose et al., 2025).

However, semantic information in hidden states is highly entangled with syntax, residual context, and modality-specific variations, many of which are irrelevant to the model's instantaneous decision (Neo et al.; Geva et al., 2021; Elhage et al., 2021). Consequently, raw hidden-state differences may be dominated by large but semantically uninformative variations. This motivates an explicit mechanism to isolate decision-relevant semantic components, enabling us to measure how visual inputs influence token prediction along meaningful directions.

We therefore treat these **Top-$K$ candidate tokens** as the *active decision set* at step $t$, whose embeddings approximate the model's instantaneous decision space. This does *not* assume that the Top-$K$ span covers the full task semantics; rather, it isolates the local directions through which visual perturbations can affect the relative logits of the currently competitive candidates. By restricting our analysis to this active set, we capture decision-relevant visual influence while significantly enhancing efficiency by avoiding redundant full-vocabulary computations. A more detailed discussion of this design choice is provided in the Appendix A.

Concretely, let $\boldsymbol{M}_K = [\boldsymbol{e}_1, \ldots, \boldsymbol{e}_K] \in \mathbb{R}^{d \times K}$ denote the output embeddings of the Top-$K$ candidates from the final decoding layer.

We define the *semantic decision subspace*

$$\mathcal{S}_t = \text{span}(\boldsymbol{M}_K), \tag{3}$$

This subspace constitutes the *Semantic Lens* at step $t$, providing a decision-aligned coordinate frame for measuring semantic influence, as shown in the left part of Fig 1.

The key representation-theoretic implication is that a hidden-state perturbation can affect the *relative ranking* among these candidates *only* through its components lying in $\mathcal{S}_t$; perturbations orthogonal to $\mathcal{S}_t$ do not change the logits restricted to this candidate set and are thus semantically inert for the current decision.

Crucially, since intermediate hidden states already reside in a partially semantically organized space, and they lie in a representational regime where cross-modal information exchange is most active, projecting them onto $\mathcal{S}_t$ provides a principled way to estimate the semantic influence of visual inputs on the model's decision process.

To obtain a stable coordinate system for this space, we com-

pute an orthonormal basis via reduced QR decomposition,

$$M_K = QR, \quad Q \in \mathbb{R}^{d \times r}, \tag{4}$$

where the columns of $Q$ form an orthonormal **semantic basis** spanning $\mathcal{S}_t$ (with $r = \mathrm{rank}(M_K)$).

Importantly, $Q$ is not used to generate tokens; rather, it serves as a *measurement basis* that disentangles decision-relevant semantic components from the full hidden representation.

This enables us to "factor out" non-semantic variations in intermediate hidden states by projecting them onto $\mathcal{S}_t$:

$$\Pi_{\mathrm{sem}} = QQ^\top. \tag{5}$$

In SIVS, $\Pi_{\mathrm{sem}}$ acts as a linear operator induced by this semantic lens, mapping hidden-state differences onto prediction-aligned semantic coordinates, thereby isolating the semantic effect attributable to visual information.

**Semantic impact induced by visual information.** The **semantic impact** is defined as

$$\boldsymbol{\delta}_{\mathrm{sem}} = \Pi_{\mathrm{sem}}\big(h_t^{(v)} - h_t^{(\emptyset)}\big). \tag{6}$$

This projection removes components orthogonal to the model's decision-relevant space, yielding a clean, prediction-aligned measure of visual semantic influence, as shown in the middle part of Fig 1.

The magnitude $\|\boldsymbol{\delta}_{\mathrm{sem}}\|$ serves as a proxy for the model's instantaneous semantic demand on visual information.

### 3.3. Part II: Token-Level Semantic Attribution

While $\boldsymbol{\delta}_{\mathrm{sem}}$ quantifies the *total* semantic influence of visual context, fine-grained KV selection requires attributing this effect to individual visual tokens. A detailed derivation of the decomposition below is provided in the Appendix B.3.

At layer $\ell^\star$, the hidden-state perturbation induced by visual information is linear in the visual value vectors:

$$h_t^{(v)} - h_t^{(\emptyset)} = \sum_{i=1}^{N} \alpha_{t,i} W_V h_{t-1}^{V_i} \triangleq \sum_{i=1}^{N} c_i, \tag{7}$$

where $\alpha_{t,i}$ denotes the attention weight and $c_i$ is the raw contribution of the $i$-th visual token.

**Semantic contribution score.** We measure how strongly each token aligns with the global semantic displacement by

$$s_i = \frac{\langle c_i, \boldsymbol{\delta}_{\mathrm{sem}} \rangle}{\|\boldsymbol{\delta}_{\mathrm{sem}}\|}. \tag{8}$$

Empirically, the distribution of $\{s_i\}$ is highly long-tailed, with a small number of tokens acting as semantic outliers.

**Adaptive selection.** Instead of selecting a fixed number of tokens, we apply a distribution-aware threshold

$$\tau_{\mathrm{kv}} = \mu_s + \alpha\sigma_s, \tag{9}$$

where $\mu_s$ and $\sigma_s$ are the mean and standard deviation of $\{s_i\}$. $\alpha$ is a tunable hyperparameter that controls the retention strictness of the dynamic threshold.

The retained visual token set is

$$\Phi_t^\star = \{i \mid s_i > \tau_{\mathrm{kv}}\}. \tag{10}$$

Visual KV pairs not in $\Phi_t^\star$ are evicted for the current $T$-step interval, and the remaining subset is reused for the next $T$ decoding steps as shown in the right part of the Fig 1.

**Controllable accuracy–compression trade-off.** The threshold coefficient $\alpha$ directly determines the operating point of SIVS: increasing $\alpha$ raises $\tau_{\mathrm{kv}}$, retains fewer visual KV pairs, and yields higher compression, while decreasing $\alpha$ preserves more visual evidence and favors accuracy. This makes adaptive selection a continuous mechanism for navigating the accuracy–compression frontier rather than a fixed-budget pruning heuristic. Empirically, varying $\alpha$ yields favorable Pareto behavior on both GQA and TextVQA, where SIVS maintains stronger accuracy under matched compression than heuristic pruning baselines. This behavior is also stable across model scales: later results on Qwen2.5-VL from 3B to 72B (Table 3) show that the same thresholding rule preserves strong operating points on both GQA and TextVQA without model-specific redesign.

**Remark: Semi-Dynamic Scheduling.** SIVS updates the selected visual KV set every $T$ decoding steps, motivated by the fact that visual dependence is temporally non-uniform. Evaluating importance at every step would increase latency and introduce noisy selections during low-vision-dependency phases. Periodic updates, therefore, focus computation on semantically critical moments while keeping selection stable. Further analysis is provided in the Sec. 4 and Appendix C.2.

## 4. Experiments

### 4.1. Experiment Setup

Our main experiments use LLaVA-1.5 7B as the base multimodal model, as most existing visual pruning methods are evaluated on this backbone, enabling direct and fair comparison. We further validate SIVS on additional architectures such as Qwen-based VLMs to demonstrate generality.

Unless otherwise specified, SIVS is applied at an intermediate layer $\ell^\star = 16$, with an update interval $T = 10$ decoding steps and adaptive coefficient $\alpha = 0.3$.

We compare SIVS with representative visual pruning methods, including FastV, SparseVLM, LOOK-M, and VisPruner. All compared methods are training-free and applied at inference time. Since VisPruner does not provide official support for multi-image inputs, we implement this component ourselves to enable evaluation on MileBench.

We evaluate SIVS on a diverse suite of multimodal benchmarks, including GQA (Hudson & Manning), MMBench (Liu et al., c), MME (Fu et al., 2026), POPE (Li et al., 2023), ScienceQA (Lu et al.), TextVQA (Singh et al.), and MileBench (Dingjie et al., 2024), covering compositional reasoning, factual verification, OCR-centric understanding, and multi-image reasoning scenarios.

### 4.2. Main Results

Table 1 reports the relative accuracy, KV compression ratio, and Table 2 shows the decoding latency. As shown in Table 1, SIVS achieves the best trade-off between accuracy and compression across all benchmarks. By removing 87.66% of visual KV states, it retains 99.93% of the vanilla model's average performance, demonstrating that semantic-impact–guided selection effectively discards redundant visual information while preserving decision-critical signals. Table 2 further shows that SIVS attains the lowest decoding latency (18.78 ms per step), yielding a 23.5% speedup over the vanilla model, confirming that reduced visual KV leads to more efficient attention computation.

SIVS delivers the largest gains on POPE and MileBench, where it slightly outperforms the vanilla model by suppressing visual distractions and handling extremely long visual contexts more effectively. On other benchmarks (MMBench, MME, ScienceQA), it matches or achieves the best performance, indicating that semantic subspace projection reliably preserves task-relevant visual evidence. Minor drops appear on GQA and TextVQA, which often rely on globally distributed cues or small textual details that may have a weak instantaneous semantic impact under aggressive compression. Nonetheless, the performance differences remain marginal.

Overall, these results show that semantic impact serves as a task-aligned criterion for visual KV scheduling, enabling near-lossless accuracy at the highest compression ratio among compared methods.

**Accuracy–Compression Trade-off.** To compare methods beyond a single operating point, we vary $\alpha$ and report the accuracy–compression Pareto frontier on GQA and TextVQA in Fig. 2. SIVS maintains stronger accuracy at high compression ratios than SparseVLM and VisPruner, indicating that its gains come from decision-aligned retention rather than a fixed pruning budget.

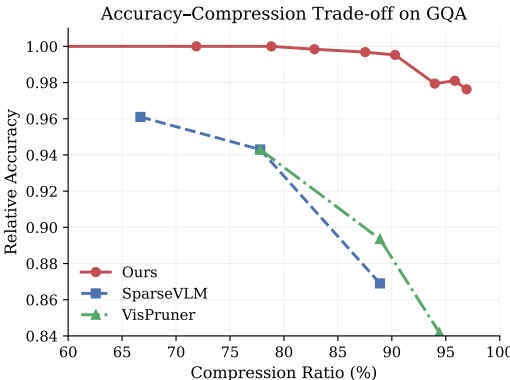

*(a)* Accuracy–compression Pareto comparison on GQA.

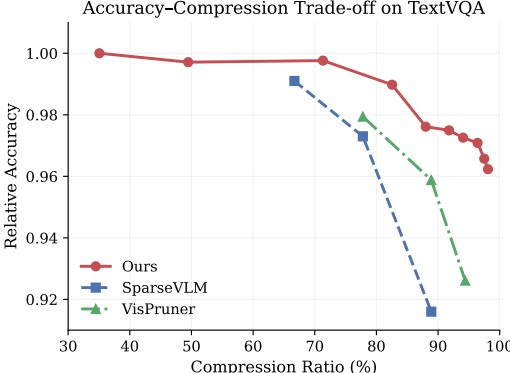

*(b)* Accuracy–compression Pareto comparison on TextVQA.

*Figure 2.* Accuracy–compression Pareto comparison. SIVS achieves a better trade-off by retaining visual KV states according to semantic impact.

**Scaling Results on Qwen2.5-VL.** To further validate the generality of SIVS beyond LLaVA-1.5, we evaluate our method on Qwen2.5-VL models from 3B to 72B. SIVS is applied without any modification, as it operates solely on decoding-stage visual KV states. We do not include comparisons with other visual compression methods in this setting, as existing baselines do not provide official implementations compatible with Qwen-based architectures, making a fair cross-method evaluation infeasible.

As shown in Table 3, SIVS maintains 86–91% visual KV compression with near-lossless performance across model scales. The results indicate that the prediction-conditioned semantic lens depends on local decision geometry rather than a specific model size, and its behavior remains stable when moving to substantially larger VLMs.

### 4.3. Ablation Studies

We conduct targeted ablations to analyze SIVS from both semantic estimation and scheduling perspectives. Experiments on TextVQA and GQA evaluate the correctness and stability

*Table 1.* Performance comparison between SIVS and alternative visual context compression methods. Acc indicates the relative performance across benchmarks.

| Method | Benchmarks | | | | | | | Acc | Compress |
| --- | --- | --- | --- | --- | --- | --- | --- | --- | --- |
| | GQA | MMB | MME | POPE | SQA | VQA$^{\text{Text}}$ | MileBench$^{\text{Real}}$ | | |
| Vanilla | 61.9 | 64.6 | 1864 | 85.9 | 69.5 | 58.3 | 38.08 | 100% | - |
| FastV (ECCV24) | 49.6 | 56.1 | 1490 | 53.4 | 68.6 | 50.5 | 37.96 | 96.46% | 77.80% |
| LOOK-M (EMNLP24) | **61.9** | 64.4 | 1857 | 86.57 | 69.1 | 56.8 | 38.56 | 99.85% | 80% |
| SparseVLM (ICML25) | 58.4 | 64.5 | 1746 | 85.0 | 68.6 | 56.7 | 36.22 | 96.84% | 77.80% |
| VisPruner (ICCV25) | 58.2 | 62.7 | 1461 | 84.6 | 69.1 | **57.0** | 38.57 | 95.20% | 77.8% |
| SIVS (Ours) | 61.7 | **64.6** | **1864** | **86.2** | **69.4** | 56.85 | **38.94** | **99.95%** | **87.66%** |

*Table 2.* Decoding step latency

| Method | Decode Step Time |
| --- | --- |
| Vanilla | 24.56ms |
| FastV | 21.62ms |
| LOOK-M | 21.96ms |
| SparseVLM | 23.34ms |
| VisPruner | 19.94ms |
| SIVS (Ours) | **18.78ms** |

*Table 3.* Scaling results on Qwen2.5-VL models.

| Dataset | Model | Vanilla | SIVS | Retention | Compress |
| --- | --- | --- | --- | --- | --- |
| GQA | 3B | 58.89 | 58.86 | 99.95% | 91.08% |
| | 7B | 60.77 | 60.76 | 99.98% | 90.87% |
| | 32B | 62.06 | 61.76 | 99.52% | 88.32% |
| | 72B | 62.16 | 62.26 | 100.16% | 87.94% |
| TextVQA | 3B | 78.17 | 77.32 | 98.91% | 88.07% |
| | 7B | 84.50 | 81.92 | 96.59% | 90.80% |
| | 32B | 79.74 | 77.70 | 97.44% | 88.94% |
| | 72B | 84.70 | 81.54 | 96.27% | 86.05% |

of semantic impact estimation for fine-grained grounding and reasoning, while LLaVA-Bench is used to study temporal scheduling and efficiency trade-offs due to its longer decoding sequences.

**Prompt setting.** LLaVA-style evaluation may include externally extracted OCR tokens in the prompt, which can reduce the model's reliance on visual inputs and obscure the effect of decoding-time visual scheduling. Unless specified otherwise, TextVQA ablations are conducted without OCR augmentation.

**Semantic Impact vs. Attention-Based Scoring.** We compare the proposed semantic impact score with attention-based heuristics, including raw attention weights and attention-weighted value norms (i.e., the norm of attention-weighted value vectors, corresponding to $\|c_i\|$ in Eq. 7), while keeping the selection pipeline unchanged.

*Table 4.* Semantic impact vs. attention-based token scoring on TextVQA.

| Scoring Method | Acc (%) |
| --- | --- |
| Attention weight | 37.16 |
| Attn-weighted value norm | 36.90 |
| Semantic impact (Ours) | 38.40 |

As shown in Table 4, attention-only criteria yield lower accuracy under the no-OCR TextVQA setting. This result suggests that attention magnitude alone is insufficient to estimate a token's contribution to the final semantic outcome, highlighting the importance of decision-aligned semantic impact estimation.

**Layer-wise Performance.** Layer-wise analysis (Fig. 3) shows that mid-to-late layers achieve near-optimal accuracy. In contrast, early layers underperform because they primarily encode low-level visual features that lack strong semantic alignment, while the deepest layers suffer from the attenuation of fine-grained visual signals after extensive multimodal fusion. This stable depth window suggests that semantic impact estimation leverages robust, shared representational properties rather than layer-specific effects, successfully balancing visual sensitivity with semantic alignment. See Appendix C.1 for details.

*Table 5.* Best layer for semantic impact estimation across model scales.

| Model | Total Layers | Best $\ell^\star$ | Ratio |
| --- | --- | --- | --- |
| Qwen2.5-VL-3B | 36 | 22 | 0.61 |
| Qwen2.5-VL-7B | 28 | 17 | 0.61 |
| Qwen2.5-VL-32B | 64 | 39 | 0.61 |
| Qwen2.5-VL-72B | 80 | 50 | 0.63 |
| LLaVA-1.5-7B | 32 | 18 | 0.56 |
| LLaVA-1.5-13B | 40 | 24 | 0.60 |

Table 5 further shows that the best-performing layer consistently falls around $0.6L$ across architectures and scales, with

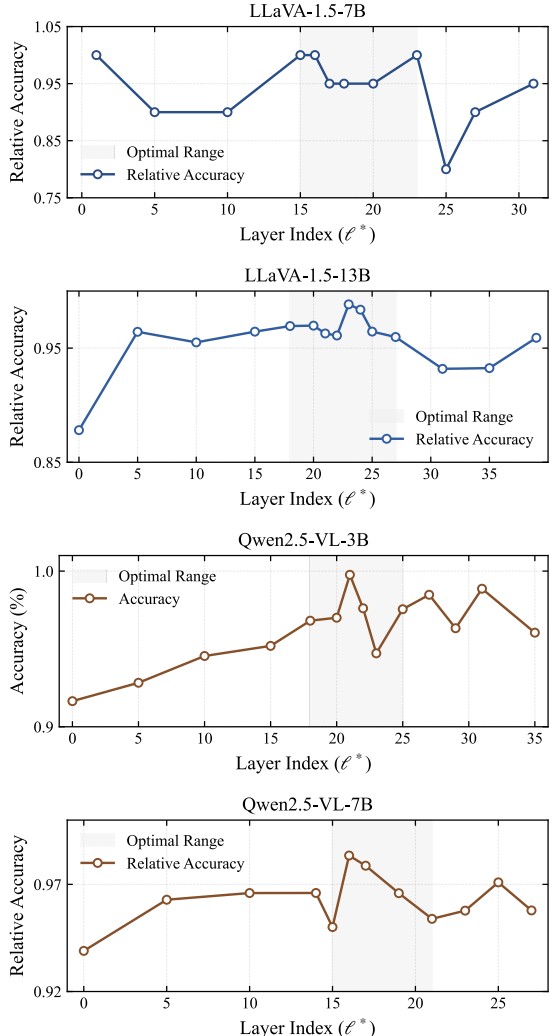

Figure 3. Effect of the selected layer $\ell^\star$ for semantic impact estimation on TextVQA. While performance is not strictly monotonic across layers, a broad range of mid-to-late layers yields comparable accuracy and efficiency.

all tested models lying in the range of $0.56L$–$0.60L$ and Qwen models concentrated within $0.61L$–$0.63L$. This indicates that SIVS depends on a relative representational stage where semantics are sufficiently formed while visual influence remains accessible, rather than on a precisely tuned absolute layer index.

**Sensitivity to Top-$K$** We vary $K$ over a wide range (Fig. 4). Performance improves as $K$ increases and stabilizes once $K$ is moderately large (e.g., $K \geq 50$), showing that a small set of high-probability candidates is sufficient to capture the dominant semantic directions. Larger $K$ mainly introduces low-probability tokens with limited influence, leading to diminishing returns.

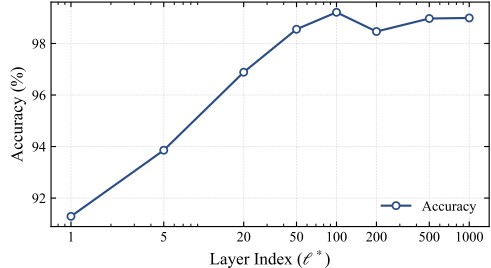

Figure 4. Effect of Top-$K$ size on performance. Accuracy improves as $K$ increases and stabilizes once $K$ is moderately large.

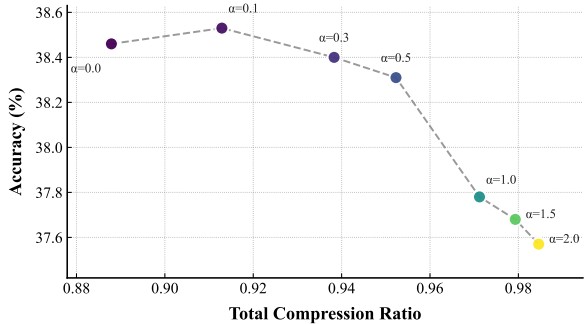

Figure 5. Accuracy–compression trade-off under different values of $\alpha$ on TextVQA.

**Impact of $\alpha$.** We study the hyperparameter $\alpha$, which controls the aggressiveness of contribution filtering. Results on TextVQA (Fig. 5) show that performance remains stable over a moderate range of $\alpha$ (e.g., 0–0.5), despite substantial increases in compression. When $\alpha$ becomes large, accuracy begins to decline as more semantically relevant tokens are removed. These results indicate that SIVS does not rely on precise threshold tuning, and that $\alpha$ provides a smooth trade-off between semantic fidelity and efficiency. This pattern indicates that semantic contribution is concentrated in a small subset of visual tokens, allowing substantial compression before essential information is affected.

**Update Interval $T$ and the Static–Dynamic Trade-off.** We study the semi-dynamic refresh interval $T$, which determines how frequently visual token selection is updated during decoding. $T \to \infty$ reduces SIVS to a static policy, while $T = 1$ performs per-step updates. We evaluate this on LLaVA-Bench, where long decoding sequences make update frequency more influential (Fig. 6).

The results show a clear trade-off between adaptivity and efficiency. Smaller $T$ improves responsiveness to changing semantic needs, but increases overhead, while the accuracy gains of the per-step updates are limited. Moderate values of $T$ achieve near-optimal accuracy with substantially lower latency. This indicates that the model's reliance on visual information does not evolve uniformly over time and that

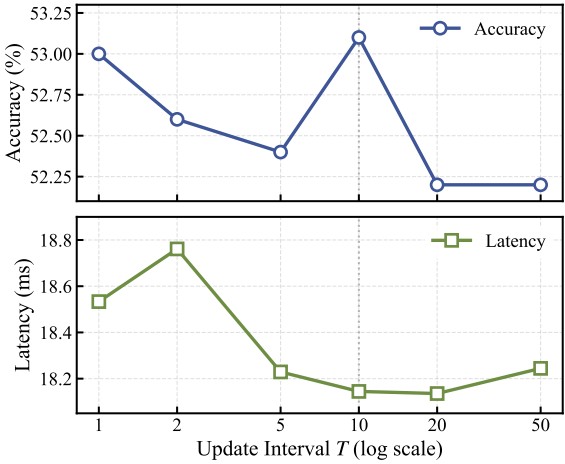

*Figure 6.* Effect of update interval $T$ on decoding accuracy and latency.

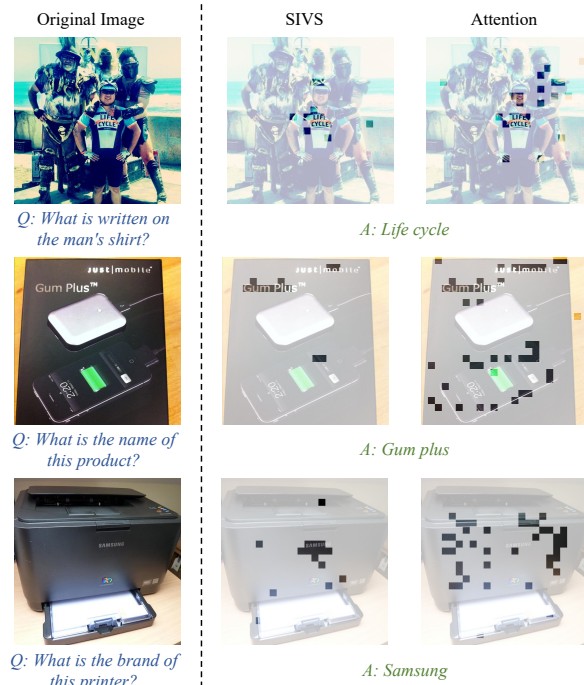

*Figure 7.* Qualitative comparison of visual KV selection. SIVS focuses on semantically decisive regions relevant to the question while suppressing scattered, task-irrelevant background activations, achieving stronger semantic alignment than attention-based selection.

periodic refresh is sufficient to capture meaningful shifts in semantic focus. A more detailed analysis of this temporal behavior is provided in the Appendix C.2. Additional analyses of long-form generation, temporal reasoning, Top-$K$ geometry, fine-grained OCR robustness, and compatibility with token merging are also provided in the Appendix.

## 5. Case Study

As illustrated in Fig. 7, where transparent patches indicate the regions selected by the model, our method concentrates visual tokens on semantically decisive regions required to answer the question, while discarding background and visually salient yet task-irrelevant content. In contrast, attention-based selection tends to spread focus over broader regions, including areas that are visually prominent but only weakly related to the final prediction. The tokens retained by our method align closely with objects and textual cues that directly influence the model's output, demonstrating that semantic impact estimation captures information that genuinely contributes to the output distribution rather than merely reflecting raw visual saliency.

## 6. Conclusion

We presented Semantic Impact–Driven Visual Scheduling (SIVS), an inference-time method that decouples visual KV retention from local attention statistics. By grounding token selection in the geometry of the semantic decision boundary, SIVS filters out visually salient yet semantically irrelevant tokens. Experiments show that SIVS improves accuracy–efficiency trade-offs over strong compression baselines while remaining compatible with system-level optimizations such as FlashAttention. More broadly, this work highlights the limitation of attention as a proxy for importance and

uses a Semantic Lens to establish semantic impact as a principled criterion for resource allocation in autoregressive MLLMs. Future work includes extending this semantic-guided paradigm to long-video understanding and hierarchical memory management.

## Acknowledgements

We would like to thank anonymous reviewers for their suggestions and comments sincerely. The work was supported by the Beijing Natural Scienece Foundation (L247010).

## Impact Statement

This paper presents work whose goal is to improve the efficiency of vision-language models and to provide a clearer understanding of how visual information contributes to their semantic decision-making and cross-modal alignment during decoding. We do not identify any new ethical concerns introduced specifically by the proposed method.

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

# A. Additional Theoretical Justification

## A.1. Why the Top-$K$ Subspace Reflects Instantaneous Semantic Decisions

At decoding step $t$, the next-token distribution is produced by a linear projection of the hidden state onto the vocabulary embedding matrix. In practice, only a small subset of tokens dominates the probability mass and competes in logit space. Let $M_K = [e_1, \ldots, e_K]$ denote the embeddings of the Top-$K$ candidate tokens. Any hidden-state perturbation $\Delta h$ affects their relative ranking only through components lying in $\mathrm{span}(M_K)$, since logits are computed via inner products $e_i^\top h$. Perturbations orthogonal to this span induce approximately uniform shifts and thus have negligible influence on token ordering.

Therefore, $\mathrm{span}(M_K)$ provides a practical approximation to the model's instantaneous semantic decision space. This claim is local rather than exhaustive: the Top-$K$ subspace is not intended to span all task semantics, but to capture directions that can affect the current competition among active next-token candidates. Visual evidence supporting a later or currently low-probability alternative may still be outside this local lens, which is one reason SIVS periodically refreshes the retained visual KV set during decoding.

This restriction is not only semantically well motivated, but also computationally advantageous. Instead of performing subspace construction over the full vocabulary embedding matrix (which can contain tens of thousands of tokens), we only need to operate on the Top-$K$ embeddings. Consequently, the QR decomposition used to build the semantic basis scales with $K$ rather than the vocabulary size $|\mathcal{V}|$, reducing both memory movement and matrix factorization cost by orders of magnitude. Since $K \ll |\mathcal{V}|$ in practice, this design makes semantic subspace estimation lightweight enough to be executed at every $T$ decoding step, enabling dynamic visual scheduling without introducing noticeable inference overhead.

## A.2. Geometry of the Top-$K$ Candidate Subspace

We further examine whether the Top-$K$ candidate embeddings collapse into a narrow or clustered subspace. For $K = 50$, the candidate embedding matrix is consistently full rank in our evaluation. To measure effective dimensionality beyond exact rank, we compute the effective rank (Roy & Vetterli, 2007). Given singular values $\{\sigma_i\}$, we normalize them as $p_i = \sigma_i / \sum_j \sigma_j$ and compute

$$\mathrm{erank}(M_K) = \exp\left(-\sum_i p_i \log p_i\right).$$

The mean effective rank is 47.17, and the minimum observed effective rank is 43.22. These values are close to the full rank of 50, indicating that the Top-$K$ candidate embeddings form a well-spread, non-degenerate subspace rather than a strongly clustered set. This supports the use of the Top-$K$ span as a stable local measurement basis for semantic impact estimation.

## A.3. Semantic Linearity of Intermediate Representations

Although the output distribution is produced at the final layer, prior work on representation linearity and logit lens analysis suggests that mid-to-late transformer layers already encode linearly decodable semantic information. These layers balance semantic abstraction with sensitivity to multimodal interactions, making them suitable for analyzing how visual inputs influence the evolving language representation. For this reason, we estimate semantic impact at an intermediate layer $\ell^\star$ rather than at the output layer.

## A.4. Interpreting Projected Hidden-State Differences as Semantic Impact

To quantify the influence of visual information on the model's decision-making, we analyze the hidden state difference $\Delta h = h^{(v)} - h^{(\emptyset)}$, which represents the perturbation induced by visual tokens. To isolate the components of this perturbation that are functionally relevant to the current prediction, we project $\Delta h$ onto a context-specific semantic subspace $\mathcal{S}$.

$M_K$ is the Top-$K$ candidate tokens embeddings. Since they are generally not orthogonal, we obtain an orthonormal basis $Q$ for $\mathrm{span}(M_K)$ (e.g., via a QR decomposition of $M_K$).

The *semantic impact vector* is then defined as

$$\delta_{\mathrm{sem}} = QQ^\top \Delta h.$$

This projection $\delta_{\mathrm{sem}}$ filters out any visual influence that is orthogonal to the subspace spanned by the most likely next tokens.

The magnitude $\|\delta_{\text{sem}}\|$ thus serves as a precise proxy for Semantic Impact: it measures how much of the visual information is aligned with the model's potential semantic outputs.

# B. Extended Methodological Details

## B.1. Semantic Subspace Construction

At each decoding step, we select the Top-$K$ tokens according to the current output logits. Their output embeddings form $M_K \in \mathbb{R}^{d \times K}$. We compute a reduced QR decomposition

$$M_K = QR,$$

where $Q \in \mathbb{R}^{d \times K}$ provides an orthonormal basis for the semantic subspace. In practice, $K = 50$ works well across models and datasets, and we observe low sensitivity to this choice (see Section A.2).

## B.2. Layer Selection

Semantic impact is estimated at an intermediate layer $\ell^\star$ that satisfies two key criteria related to representation maturity and multimodal interaction.

First, the hidden representations at this layer should already encode a relatively stable and linearly decodable semantic structure. In very shallow layers, activations are dominated by modality-specific processing and low-level feature transformation, and the emerging semantics are not yet well aligned with the model's prediction space. Estimating semantic impact at such depths would therefore mix semantic effects with unresolved perceptual or lexical processing.

Second, cross-modal interactions should still be active at this layer. In very deep layers, multimodal fusion has largely been completed, and subsequent transformations focus more on consolidating semantic decisions and refining syntactic structure in the language stream. At that stage, visual influence has already been substantially integrated, making it harder to disentangle how visual inputs are currently steering the evolving representation.

Mid-to-late layers naturally satisfy both conditions: semantics has largely formed and can be meaningfully projected into a prediction-conditioned semantic subspace, while visual–textual interactions remain sufficiently explicit to expose the ongoing influence of visual context. We therefore estimate semantic impact at a representative intermediate layer $\ell^\star$. Empirical layer-wise statistics and performance trends are provided in Section C.1.

## B.3. Token-Level Semantic Impact Decomposition

While $\delta_{\text{sem}}$ captures the *total* semantic influence of visual context at a decoding step, practical visual KV selection requires attributing this effect to individual visual tokens.

**From cross-attention to visual-induced hidden-state updates.** At layer $\ell^\star$, visual information influences the language hidden state through a standard cross-attention mechanism. Let the query vector of the current language token be $q_t$, and let $\{k_i, v_i\}_{i=1}^N$ denote the key and value vectors of the visual tokens. The cross-attention output is

$$\text{Attn}(q_t, K, V) = \sum_{i=1}^N \alpha_{t,i}\, v_i, \qquad \alpha_{t,i} = \text{softmax}\left(\frac{q_t^\top k_i}{\sqrt{d}}\right) \tag{11}$$

where $\alpha_{t,i}$ is the attention weight assigned to visual token $i$. Since the value vectors are linear projections of visual hidden states, $v_i = W_V h_{t-1}^{V_i}$, the attention output — and hence the visual contribution to the updated language representation — is a weighted sum of linearly transformed visual features.

Consequently, the hidden-state perturbation induced by visual inputs can be written as

$$\Delta h = h_t^{(v)} - h_t^{(\emptyset)} = \sum_{i=1}^N \alpha_{t,i}\, W_V h_{t-1}^{V_i} \triangleq \sum_{i=1}^N c_i \tag{12}$$

where $\Delta h_i$ denotes the hidden-state update attributable to the $i$-th visual token.

**Projection into the semantic subspace.** Recall that the semantic impact vector is defined as the projection of $\Delta\boldsymbol{h}$ onto the prediction-conditioned semantic subspace:

$$\delta_{\text{sem}} = \boldsymbol{Q}\boldsymbol{Q}^\top \Delta\boldsymbol{h} \tag{13}$$

Substituting the linear decomposition above gives

$$\delta_{\text{sem}} = \sum_{i=1}^{N} \boldsymbol{Q}\boldsymbol{Q}^\top \boldsymbol{c}_i \tag{14}$$

showing that each visual token contributes additively to the final semantic displacement within the decision-relevant subspace. We therefore define the *token-level semantic contribution* of visual token $i$ as

$$s_i = \frac{\langle \boldsymbol{c}_i, \boldsymbol{\delta}_{\text{sem}} \rangle}{\|\boldsymbol{\delta}_{\text{sem}}\|} \tag{15}$$

which measures how strongly the perturbation induced by token $i$ aligns with the overall semantic correction direction. Tokens with larger $c_i$ values are those whose visual information most directly supports the model's current semantic decision, whereas tokens with near-zero or negative contributions have limited or misaligned semantic influence.

## C. Extended Ablation Studies

### C.1. Layer-wise Statistics of $\delta_{\text{sem}}$

We report the layer-wise distribution of $\|\delta_{\text{sem}}\|$ and discuss its relation to the choice of the intermediate layer $\ell^\star$, shown in Fig 8 and Fig 9 Across models, $\|\delta_{\text{sem}}\|$ generally increases from early to deeper layers, indicating that deeper hidden representations admit larger projection onto the prediction-conditioned semantic subspace.

However, we empirically observe that the layer achieving the largest (or fastest-growing) $\|\delta_{\text{sem}}\|$ does *not* necessarily coincide with the layer that yields the best downstream performance under SIVS-based scheduling.

Across the scaling experiments, the best layer consistently appears near a fixed relative depth rather than a fixed absolute index: Qwen-3B, Qwen-7B, Qwen-32B, and Qwen-72B peak around $0.61L$–$0.63L$, while LLaVA-7B and LLaVA-13B peak at $0.56L$ and $0.60L$, respectively. This pattern supports using a coarse mid-to-late layer band for $\ell^\star$ selection. This

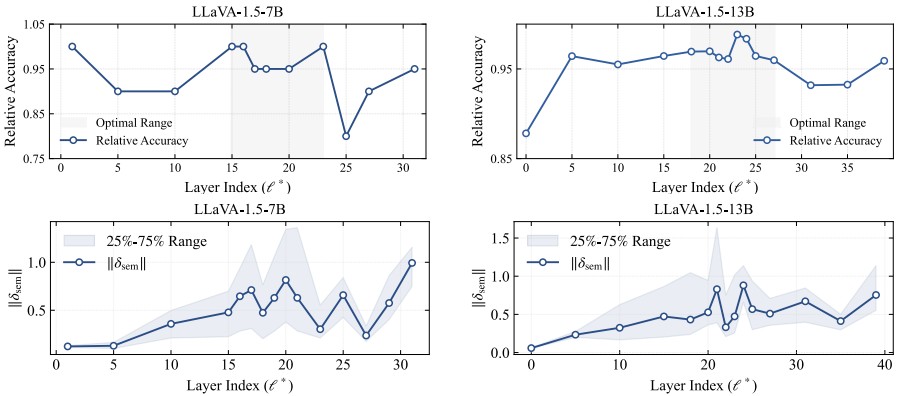

*Figure 8.* Layer-wise statistics of semantic impact magnitude $|\delta_{\text{sem}}|$ aggregated over the full evaluation set. The curve shows the median value at each layer, while the shaded region indicates the interquartile range (25%–75%), illustrating how the strength and stability of semantic impact vary across network depth.

mismatch is expected for two reasons. First, $\|\delta_{\text{sem}}\|$ measures the *magnitude* of projected semantic impact, but does not distinguish between *decision-relevant* semantic change and amplified but unstable variations. In very late layers, hidden states can become highly sensitive to small perturbations due to strong language-side refinement and logit-level sharpening; consequently, $\|\delta_{\text{sem}}\|$ may increase while the corresponding selection signal becomes noisier and less transferable to robust visual token retention. Second, the effectiveness of $\ell^\star$ depends on a trade-off between (i) semantic abstraction (too early layers under-represent decision semantics) and (ii) stability/conditioning of the projection (too late layers may exhibit higher

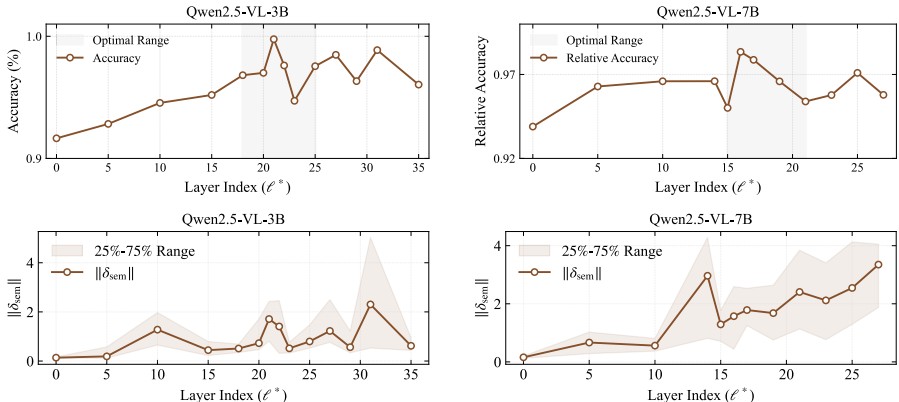

*Figure 9.* Layer-wise statistics of semantic impact magnitude $|\delta_{\text{sem}}|$ aggregated over the full evaluation set. The curve shows the median value at each layer, while the shaded region indicates the interquartile range (25%–75%), illustrating how the strength and stability of semantic impact vary across network depth.

variance and weaker locality of visual-to-text attribution), such that an intermediate range can be optimal even when it does not maximize $\|\delta_{\text{sem}}\|$. Consistent with this interpretation, mid-to-late layers often show *more stable* $\|\delta_{\text{sem}}\|$ distributions (e.g., tighter interquartile ranges) compared to layers where $\|\delta_{\text{sem}}\|$ spikes or fluctuates sharply. Therefore, layer-wise statistics of $\|\delta_{\text{sem}}\|$ should be viewed as a *diagnostic* that reveals where semantic-impact signals emerge and stabilize, rather than a direct proxy for selecting $\ell^\star$ by maximizing $\|\delta_{\text{sem}}\|$.

In practice, we choose $\ell^\star$ within a contiguous mid-to-late layer band where performance is consistently strong and $\|\delta_{\text{sem}}\|$ exhibits moderate magnitude with improved stability.

## C.2. Temporal Evolution of Semantic Impact

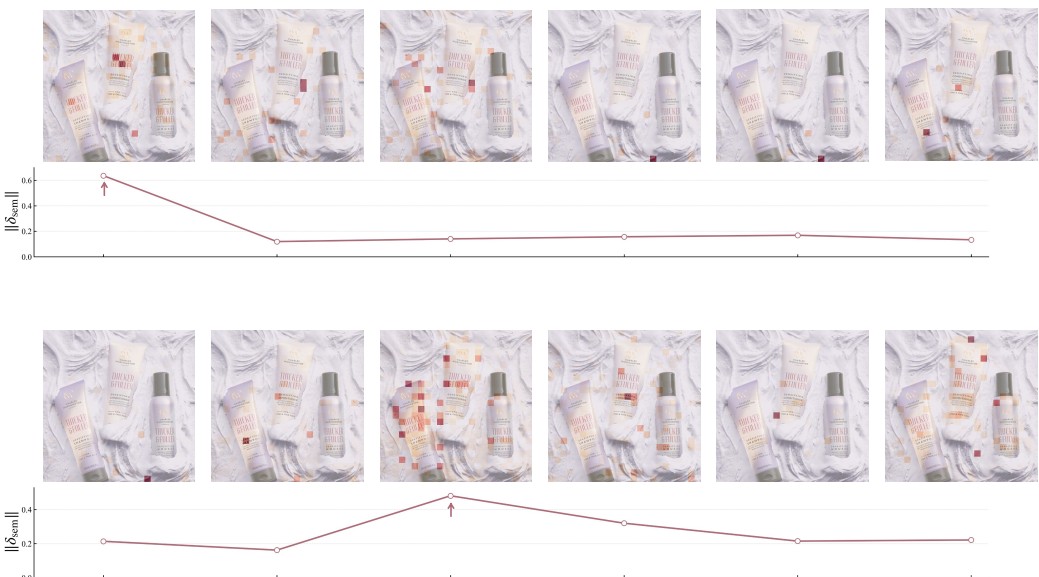

*Figure 10.* Temporal evolution of semantic demand during decoding. The magnitude $\|\boldsymbol{\delta}_{\text{sem}}\|$ exhibits sparse, bursty behavior, indicating that strong visual dependence arises only at a few semantically decisive steps.

Figure 10 reveals a clear temporal structure in how the model utilizes visual information during autoregressive decoding. Rather than continuously relying on visual inputs at every step, the model exhibits bursty semantic dependence: a small number of decoding steps show large semantic displacement $|\boldsymbol{\delta}_{\text{sem}}|$, while most steps remain in a low-impact regime. This behavior can be understood from the perspective of information flow through the attention mechanism.

At steps where $|\delta_{\text{sem}}|$ is high, the model actively queries visual tokens to resolve semantically decisive content (e.g., object identity, attributes, or spatial relations). The retrieved visual evidence is then integrated into the hidden state and subsequently propagated through the self-attention stack into the textual key–value cache. In later steps, the model can continue reasoning using these internally stored representations, without directly attending to the image again. As a result, visual attention becomes more diffuse and less semantically targeted, leading to a gradual decay in $|\delta_{\text{sem}}|$ after each peak.

Importantly, during these low-impact phases, the model is no longer strongly conditioned on fresh visual evidence. Any attempt to perform fine-grained visual token selection at such steps becomes unreliable: the induced hidden-state difference is small and dominated by non-semantic variations, making the estimated visual importance noisy. This explains why per-step visual re-evaluation is both computationally wasteful and statistically unstable when $|\delta_{\text{sem}}|$ is low.

These observations motivate our semi-dynamic scheduling strategy. By refreshing the selected visual tokens only at periodic intervals, the method aligns visual token updates with phases of genuine semantic demand. This design achieves two complementary benefits: (1) Efficiency — avoiding unnecessary recomputation when the model is primarily relying on previously integrated visual information stored in the textual KV cache; (2) Effectiveness — performing visual token selection only when the semantic signal is strong, resulting in more reliable identification of task-relevant visual regions.

Therefore, the periodic evolution pattern in Figure 10 provides empirical support for the core assumption behind semi-dynamic visual scheduling: visual grounding in MLLMs is temporally sparse and state-dependent, rather than uniformly required at every decoding step.

### C.3. Long-Form and Temporal Reasoning

We evaluate SIVS on LLaVA-Bench to test long-form generation, where responses are much longer than typical VQA answers. The average output length is 138 tokens, with a maximum length of 881 tokens. As shown in Table 6, SIVS preserves 98.89% of the vanilla score while compressing 89.78% of visual KV states, suggesting that periodic visual scheduling remains stable over extended decoding.

*Table 6.* Long-form generation results on LLaVA-Bench.

| Method | Score | Retention | Compress |
|---|---|---|---|
| Vanilla | 45.0 | 100% | 0% |
| SIVS (Ours) | 44.5 | 98.89% | 89.78% |

MileBench further evaluates multi-image and temporally structured reasoning. Table 7 shows that SIVS remains comparable to the vanilla model across temporal categories and improves counterfactual/state-change reasoning, indicating that the retained visual KV states preserve temporally relevant evidence.

*Table 7.* Temporal reasoning results on MileBench.

| Task Category | LLaVA-7B | SIVS |
|---|---|---|
| T-1 Action Understanding | 39.33 | 39.33 |
| T-2 Object/Scene Understanding | 46.00 | 46.12 |
| T-3 Navigation & Localization | 32.25 | 31.75 |
| T-4 Counterfactual & State Change | 38.88 | 41.88 |

### C.4. Compatibility with Token Merging

SIVS is complementary to token merging because it operates at a different stage. Token merging reduces visual redundancy in the vision encoder, while SIVS performs decision-conditioned visual KV pruning during LLM decoding. We combine SIVS with ToME (Bolya et al., 2023) and report the result in Table 8. The combined pipeline keeps the benefit of encoder-side merging and further improves the dynamic decoding-stage compression.

*Table 8.* Compatibility between ToMe and SIVS on GQA.

| Method | GQA | Compress |
|---|---|---|
| Vanilla | 61.9 | 0 |
| SIVS | 61.7 | 90.28% |
| ToMe | 58.7 | 63.9% |
| SIVS+ToMe | 59.8 | 84.99% |

## D. Visualization and Case Studies

### D.1. Qualitative Token Selection Examples

We visualize the visual tokens retained by SIVS and compare them with those selected using attention-based importance, as shown in Fig. 11. Across diverse scenes, raw attention maps tend to be spatially dispersed, often highlighting broad regions that include background textures or visually salient yet semantically irrelevant areas. Using the attention-weighted value magnitude produces more concentrated activations and generally retains more semantically meaningful content; however, it still preserves a considerable amount of noisy or redundant regions due to its reliance on local interaction strength rather than decision relevance. In contrast, SIVS produces compact and structured selections that consistently align with task-relevant elements such as textual cues, foreground objects, and discriminative visual evidence, while suppressing scattered background responses. This qualitative behavior supports our claim that semantic impact—measured by contribution to the model's prediction space—more accurately reflects the model's true reliance on visual information during decoding, effectively combining the strengths of attention localization and value-aware filtering while reducing noise.

### D.2. Fine-Grained OCR Robustness

To assess whether SIVS disproportionately removes small textual cues, we group TextVQA examples by the relative area of OCR regions and compare vanilla performance with SIVS in Table 9. SIVS shows only marginal degradation even in the smallest-region bins, suggesting that semantic-impact selection remains robust when task-relevant evidence is visually small.

*Table 9.* TextVQA performance grouped by OCR region area ratio.

| Area Ratio | Count | Vanilla | SIVS |
|---|---|---|---|
| $[0, 0.0025)$ | 417 | 0.7906 | 0.7899 |
| $[0.0025, 0.005)$ | 448 | 0.7674 | 0.7571 |
| $[0.005, 0.01)$ | 414 | 0.7896 | 0.7872 |
| $[0.01, 0.025)$ | 410 | 0.7951 | 0.7893 |
| $[0.025, 1.0]$ | 453 | 0.7466 | 0.7444 |

### D.3. Failure Cases

Despite its overall precision, SIVS can fail when semantically critical information is visually subtle or only weakly aligned with the dominant prediction-conditioned semantic directions, as shown in Fig. 12. In practice, failure cases frequently arise in visually cluttered scenes or tasks that require fine-grained spatial reasoning, such as identifying small text, distinguishing subtle size differences, or locating objects based on relative position, scale, or layout. In these scenarios, the model's intermediate representation may not form a sharply concentrated semantic direction tied to the truly decisive evidence. Instead, the prediction space can reflect a mixture of multiple plausible visual cues, causing the semantic displacement induced by the actually relevant regions to be diluted. As a result, SIVS may either retain overly broad regions or prune small but crucial details.

An additional factor is that these examples are drawn from VQA-style tasks, where target answers are typically short, and decoding often terminates quickly. The model may produce an answer before a detailed reasoning trajectory—such as precise spatial localization or step-by-step visual disambiguation—has fully emerged in its hidden states. Because SIVS measures semantic impact with respect to the model's instantaneous prediction space, early-stage predictions that rely on coarse or

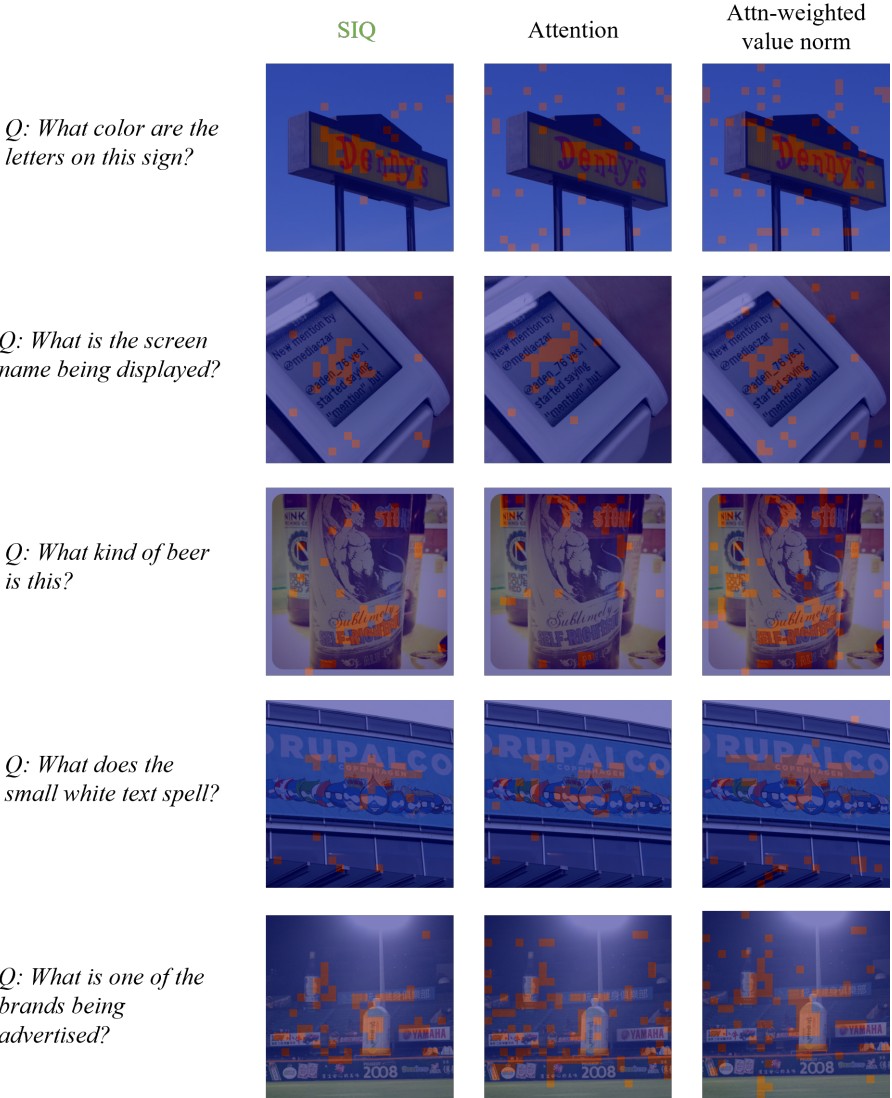

*Figure 11.* Qualitative comparison of visual KV selection. Compared with attention-based selection, SIVS concentrates tokens on semantically task-relevant regions (e.g., text, objects, and key entities) while substantially reducing scattered background activations.

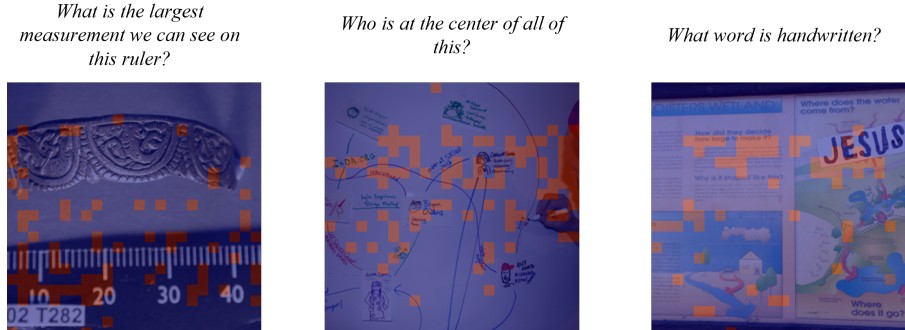

*Figure 12.* Failure cases of SIVS-based visual token selection. When task-relevant cues require fine-grained spatial reasoning, SIVS may underestimate their semantic impact and discard them, leading to missed evidence.

heuristic cues can bias the estimated semantic directions, limiting the method's ability to highlight fine-grained evidence. We hypothesize that under longer reasoning chains or tasks requiring more extended deliberation, later decoding steps could yield more stable and discriminative semantic subspaces, allowing SIVS to progressively refine visual localization and better capture subtle yet decisive cues.

More generally, SIVS can underestimate key evidence when the decisive visual signal is extremely small, when the task requires precise spatial relations, or when the model's vision-language alignment is weak at the selected layer. This limitation reflects a common trade-off in high-compression visual context methods: aggressively reducing visual KV can remove faint but critical cues, even when most redundant context is safely discarded.

