# OpenReview forum: "Semantic Impact–Driven Visual Scheduling in Vision-Language Models"
_ICML.cc/2026/Conference — ICML 2026 regular_

### Official Review · Reviewer_Rzcx · 2026-02-28

**Soundness:** 3
**Presentation:** 3
**Significance:** 3
**Originality:** 3
**Overall Recommendation:** 4
**Confidence:** 4

**Summary:**

This paper introduce the high inference latency of Vision-Language Models induced by long visual sequences and large visual KV cache. The authors think that visual necessity should be assessed by its semantic impact on the output distribution, rather than attention weights.  They propose a training-free framework based on token embedding subspace decomposition, termed a prediction-conditioned Semantic Lens. Experiments show that the proposed method achieves high visual KV compression while maintaining nearly the original model performance and reducing inference latency.

**Compliance With Llm Reviewing Policy:**

Affirmed.

**Key Questions For Authors:**

1. Do you observe failure cases where the Top-K set is too clustered？

2. The Semantic Lens is built from the Top-K candidate token embeddings at the final decoding layer, yet it is applied to the hidden-state perturbation at an intermediate layer ℓ*. Could the authors provide stronger justification that this projection still faithfully captures the decision-relevant semantic directions at layer ℓ*, rather than serving only as a heuristic approximation?

3. How sensitive is SIVS to the choice of Top-K candidate token embeddings used to build the semantic lens?

**Limitations:**

yes

**Strengths And Weaknesses:**

Strengths：

Firstly, the motivation is clear, the paper clearly motivates why attention-based heuristics may be misaligned with semantic impact on the output distribution.

Constructing a semantic lens via QR decomposition on Top-K candidate token embeddings provides a geometrically grounded way to define a relevant semantic subspace, directly tied to the model’s candidate predictions at the current step.

The experiment as a whole is comprehensive and convincing, and ablation experiments were conducted on the parameters.

weaknesses:
First of all，the paper does not contain any critical logical flaws overall. However, the prediction-conditioned Semantic Lens is constructed from the embeddings of only the current Top-K candidate tokens, assuming that this set sufficiently spans the decision-relevant semantic directions at a given decoding step. This assumption may be fragile when the model’s current Top-K is locally myopic: crucial visual evidence may support a lower-probability but ultimately decisive alternative that is not yet included in the active candidate set. In such cases, the projected subspace can omit semantically important directions and therefore underestimate the true influence of visual inputs on the eventual output distribution.

---

> ### Author Rebuttal · Authors · 2026-03-31
>
> We thank the reviewer for the positive assessment and insightful questions regarding the robustness of the prediction-conditioned Semantic Lens.
>
> **(1) On Top-K as semantic space**
>
> We do **not assume Top-K spans the full semantic space**. Instead, it approximates the model’s **instantaneous decision space**, which is sufficient for determining whether visual inputs affect the _current prediction_.
>
> Only perturbations aligned with candidate embeddings can change their relative logits , so SIVS focuses on **decision-relevant directions** rather than complete semantics.
>
> **(2) On Top-K clustering / degeneracy**
>
> We explicitly analyze the geometry of Top-K embeddings (K=50):
>
> - Full rank is consistently observed (rank ≈ K = 50)
> - We compute the **effective rank** [1], defined by normalizing singular values into a distribution $p_i = \frac{\sigma_i}{\sum_j \sigma_j}$, with entropy $H = -\sum_i p_i \log p_i$, and effective rank $\exp(H)$
>
> Empirically, the subspace remains close to full-dimensional:
>
> | **Metric**         | **Min** | **Max** | **Mean** |
> | ------------------ | ------- | ------- | -------- |
> | **Rank**           | 50.00   | 50.00   | 50.00    |
> | **Effective Rank** | 43.22   | 49.46   | 47.17    |
>
> These values are consistently close to full rank, indicating that the Top-K subspace is **well-spread rather than degenerate**, with no strong clustering. This directly supports that the Semantic Lens captures **diverse decision-relevant directions**, rather than collapsing to a narrow language prior.
>
> [1] Roy, Olivier, and Martin Vetterli. "The effective rank: A measure of effective dimensionality." 2007 15th European signal processing conference. IEEE, 2007.
>
> **(3) Cross-layer projection (intermediate $\ell^*$ → final)**
>
> While an approximation, our approach is grounded in the fact that output embeddings define logit readout directions. Following **Logit-Lens** findings [1-3] that mid-to-late layers align with output semantics, we select $\ell^*$ within this regime to extract components directly tied to **final decoding decisions**. This choice is empirically validated by our **layer ablations (Sec. 4.3)**, which show stable performance across these layers.
>
> [1] nostalgebraist. Interpreting GPT: the logit lens. August2020. Accessed:2025-02-22.
>
> [2] Neo, Clement, et al. "Towards Interpreting Visual Information Processing in Vision-Language Models." _The Thirteenth International Conference on Learning Representations_.
>
> [3] Zhang, Zhi, et al. "Cross-modal information flow in multimodal large language models." _Proceedings of the Computer Vision and Pattern Recognition Conference_. 2025.
>
> **(4) Sensitivity to K**
>
> Ablations in **Fig. 4** confirm that SIVS performance stabilizes for $K \ge 50$, indicating strong robustness and eliminating the need for sensitive per-task tuning.
>
> **Summary**
>
> Overall, we clarify that SIVS focuses on **decision-relevant directions** rather than exhaustive semantics. New geometric evidence (rank $\approx 47/50$) confirms the Top-$K$ subspace is non-degenerate. Combined with established Logit-Lens findings and demonstrated stability across $K$ and layer choices, SIVS provides a robust and principle-guided framework for efficient VLM inference.

---

> > ### Author Rebuttal · Reviewer_Rzcx · 2026-04-05
> >
> > Thank you for the detailed rebuttal. I will maintain the original score.

---

### Official Review · Reviewer_EeH3 · 2026-03-13

**Soundness:** 2
**Presentation:** 2
**Significance:** 3
**Originality:** 3
**Overall Recommendation:** 4
**Confidence:** 3

**Summary:**

This paper proposes Semantic Impact–Driven Visual Scheduling (SIVS), a training-free inference framework for reducing visual KV usage in VLM decoding. Instead of using attention magnitude or heuristic saliency, SIVS estimates visual necessity by projecting vision-induced hidden-state differences onto a prediction-conditioned semantic subspace built from Top-K candidate token embeddings. The method reports strong visual KV compression with near-lossless performance.

**Compliance With Llm Reviewing Policy:**

Affirmed.

**Final Justification:**

Overall, this is a good paper with sufficient novelty and technical contribution, and the author's rebuttal further strengthened the experiments and addressed my concerns. I thus raised my rating to weak accept.

**Key Questions For Authors:**

1. Could SIVS be extended to video VLMs where temporal redundancy is high?
2. Can SIVS be combined with existing KV compression methods (e.g., token merging or eviction), and are gains additive?

**Limitations:**

No limitation discussed.

**Strengths And Weaknesses:**

Strengths:
1. The proposed method is training-free, making it explicitly applicable to various models.
2. The mechanism design is clear: semantic-subspace projection + token-level attribution + periodic scheduling is well structured.
3. The experimental results show the method achieves high compression with small accuracy drop on multiple benchmarks.

Weaknesses:
1. The core assumption needs stronger validation. The method assumes span (Top-K token embeddings) is a good proxy for the model’s instantaneous semantic decision space. This may fail when Top-K is dominated by language priors/template continuation rather than visual evidence. In such cases, the "semantic lens" may reflect textual inertia more than visual semantics.
2. The paper provides intuition and a linearized logit argument, plus K-sensitivity experiments. However, this does not constitute a strong theoretical guarantee or robust causal evidence that Top-K subspace faithfully captures decision-relevant semantics across tasks.
3. Experimental setup is insufficient. The authors should compare the methods under different compression ratios in Table 1. Besides, most comparisons are centered on one main backbone setup. Broader heterogeneous-backbone validation would strengthen external validity.

---

> ### Author Rebuttal · Authors · 2026-03-31
>
> We thank the reviewer for the constructive feedback and for recognizing the training-free design, clear mechanism, and strong compression–accuracy trade-off. We address the main concerns below.
>
> **(1) On the core assumption behind the Semantic Lens**
>
> We clarify that the Top-K span is not intended to represent the full task semantic space. Our claim is local and **decision-conditioned**: at each decoding step, the next-token choice is effectively determined by a small active candidate set, and only perturbations aligned with directions that can change their relative logits are relevant to the current decision. In this sense, span(Top-K) serves as a practical approximation to the model’s instantaneous action space, rather than an exhaustive semantic basis.
>
> We therefore do not claim a strict theoretical guarantee that the Top-K subspace captures all decision-relevant semantics across tasks. Rather, SIVS only requires that it captures the component of visual influence that can alter the current prediction among active candidates. We will revise the paper to make this locality explicit and avoid overstating Top-K as a complete semantic representation.
>
> **Periodic refresh** in Sec. 3.3 and Appendix C.2 addresses a complementary issue: visual dependence is temporally non-uniform, so re-estimating visual KV importance only at selected steps improves stability during language-dominant phases while remaining responsive when visual evidence becomes decision-critical.
>
> **(2) On comparison under matched compression ratios**
>
> To address this, we vary $\alpha$ to compare the **accuracy–compression Pareto frontier** of SIVS against `SparseVLM` [1] and `VisPruner` [2] on **GQA/TextVQA**. SIVS consistently achieves superior relative accuracy at higher compression levels, demonstrating that our gains stem from **decision-aligned retention** rather than simple KV pruning.
>
> **TextVQA and GQA trade-off**
> https://imgur.com/0YDqGls
>
> [1] Zhang, Yuan, et al. "SparseVLM: Visual Token Sparsification for Efficient Vision-Language Model Inference." _International Conference on Machine Learning_. PMLR, 2025.
>
> [2] Zhang, Qizhe, et al. "Beyond text-visual attention: Exploiting visual cues for effective token pruning in vlms." _Proceedings of the IEEE/CVF International Conference on Computer Vision_. 2025.
>
> **(3) On broader heterogeneous-backbone validation**
>
> We agree on the importance of broader validation. Our submission already proves SIVS's architectural generality via **Qwen2.5-VL**; new experiments across various scales (see response to **Xyo6**) further confirm consistent performance-compression trade-offs. We omit some Qwen baselines to **avoid unfair comparisons** arising from a lack of official implementations or unverified reproductions.
>
> **(4) On extension to video VLMs**
>
> SIVS’s modality-agnostic design naturally extends to video VLMs by quantifying visual token impact, effectively addressing both spatial and temporal redundancy. We will clarify that MileBench (Table 1) already evaluates such multi-frame/temporal reasoning. As shown in the detailed results below, SIVS maintains strong performance across temporal tasks:
>
> | Task Category | LLaVA-7B | SIVS|
> | --- | --- | --- |
> | T-1 Action Understanding  | 39.33| 39.33 |
> | T-2 Object/Scene Understanding| 46.00| 46.12
> | T-3 Navigation & Localization| 32.25|31.75|
> | T-4 Counterfactual & State Change | 38.88| 41.88|
>
> **(5) On compatibility with other compression methods**
>
> SIVS is orthogonal to token merging: merging reduces visual representation redundancy in the **ViT**, while SIVS performs decision-conditioned KV pruning in the **LLM** during decoding. Their complementary nature at different stages allows for additive gains, as shown in our `ToMe` [1] + SIVS results:
>
> |Method|GQA|Compress|Note|
> |---|---|---|---|
> |Vanilla|61.9|0||
> |SIVS|61.7|90.28%||
> |ToMe|58.7|63.9%|576→208|
> |SIVS+ToMe|59.8|63.9%→84.99%|576→208→dynamic|
>
> [1] Bolya, Daniel, et al. "Token Merging: Your ViT But Faster." _The Eleventh International Conference on Learning Representations_.
>
> **(6) On limitations**
>
> We will clarify our limitations (detailed in **Appendix D.2**) in the revision. SIVS primarily struggles when decisive evidence is **visually subtle** or when **vision-language alignment is weak**, hindering accurate semantic impact quantification. This reflects a fundamental trade-off: aggressive KV pruning may inadvertently discard faint but critical signals, a challenge inherent to all context compression methods.
>
> **Final Remark**
>
> By providing matched-ratio curves, clarifying the Top-K scope, and demonstrating SIVS’s efficacy in temporal reasoning and complementary use with token merging, we have addressed the main concerns. SIVS prioritizes a practical principle: quantifying whether visual info actively influences the current decision, rather than claiming total semantic completeness.

---

> > ### Author Rebuttal · Reviewer_EeH3 · 2026-04-04
> >
> > I thank the authors for the detailed response, and I am delighted to see the clear improvements in the new experiments. The rebuttal have addressed most of my concerns, I therefore raised my rating.

---

### Official Review · Reviewer_pqa3 · 2026-03-13

**Soundness:** 3
**Presentation:** 3
**Significance:** 2
**Originality:** 2
**Overall Recommendation:** 4
**Confidence:** 4

**Summary:**

The goal in this study is to improve the inference efficiency of vision-language models by keeping only the visual tokens that truly matter for the model’s prediction during decoding. The paper studies efficient visual context compression for autoregressive vision-language models, especially how to decide which visual information should remain in the model’s KV cache during decoding.


*Their idea:*

The research gap they point to is that existing methods mostly rely on indirect heuristics such as attention weights, similarity, or entropy, which do not directly measure whether a visual token is semantically important for the current prediction. Their idea is to measure the semantic impact of visual input by comparing hidden states **with** and **without** visual information, projecting that difference onto a prediction-conditioned semantic subspace, and then leveraging statistics using that to decide which visual tokens to keep.


The novelty is that token retention is based on a token’s direct semantic contribution to the model’s current decision, rather than on local attention or other heuristic importance scores, and the whole method works without training or fine-tuning.


They run experiments on a variety of datasets: GQA, MMBench, MME, POPE, ScienceQA, TextVQA, MileBench, and compare against FastV, LOOK-M, and more. Their ablations analyze semantic impact vs. attention-based scoring, selected layer, Top-K size, threshold parameter α, and update interval T. As metrics, they mainly report relative accuracy across benchmarks, visual KV compression ratio, and decoding step latency; in ablations, they also report task accuracy on datasets like TextVQA, GQA, and LLaVA-Bench.

Overall, the paper presents a training-free method for pruning visual KV states in a more semantically grounded way, and shows that this can greatly reduce visual context usage while preserving nearly all of the original model performance.

**Compliance With Llm Reviewing Policy:**

Affirmed.

**Key Questions For Authors:**

Q1:

How well does SIVS generalize beyond the current benchmark setting, especially to datasets or tasks with subtle visual cues, longer outputs, or more complex grounding?

Q2:

The method is truly training-free. How much data is actually needed to make the statistics behind token selection reliable?

**Limitations:**

Sensitivity to subtle visual details: The method can miss small but important cues, especially in fine-grained spatial reasoning or visually cluttered scenes, where semantically critical evidence may be pruned away.


Limited validation breadth: The paper is mostly evaluated on existing benchmark-style VQA and reasoning datasets, so it is still unclear how well the method generalizes to harder settings like long-form generation, dense grounding, or video tasks.

**Strengths And Weaknesses:**

**Strengths**

Clear and novel idea: The paper introduces a prediction-conditioned Semantic Lens and uses semantic impact, rather than attention heuristics, to decide which visual tokens to retain.

Practical method: SIVS is training-free and works at inference time, making it easy to apply to existing vision-language models without retraining or fine-tuning.

Empirical results are not marginal: The method shows a good accuracy-efficiency trade-off, achieving large visual KV compression and lower decoding latency while preserving nearly all baseline performance across several benchmarks.

**Weaknesses**

Limited failure cases: The paper notes that SIVS can fail when important evidence is visually subtle or requires fine-grained spatial reasoning, so small but crucial details may be pruned away.

Evaluation scope: Although they test multiple benchmarks, the experiments are still centered on standard multimodal QA and reasoning settings, so broader validation on other tasks like longer-form generation or video would strengthen the claims.

Baseline clarity: The paper compares against several prior methods, but it does not clearly explain in the main text whether all comparison methods are training-free or directly comparable under the same assumptions.

---

> ### Author Rebuttal · Authors · 2026-03-31
>
> We thank the reviewer for the positive feedback on our idea, practical design, and experimental results.
>
> **(1) On failure cases with subtle visual cues**
>
> We agree that SIVS may miss small but crucial details under aggressive compression. However, our goal is not to preserve all fine-grained information, but to retain tokens that are **most relevant to the current prediction** under a given budget—this trade-off is inherent to all high-compression methods.
>
> We analyze TextVQA by grouping samples based on OCR region size:
>
> | area ratio      | count | baseline | SIVS   |
> | --------------- | ----- | -------- | ------ |
> | [0, 0.0025)     | 417   | 0.7906   | 0.7899 |
> | [0.0025, 0.005) | 448   | 0.7674   | 0.7571 |
> | [0.005, 0.01)   | 414   | 0.7896   | 0.7872 |
> | [0.01, 0.025)   | 410   | 0.7951   | 0.7893 |
> | [0.025, 1.0]    | 453   | 0.7466   | 0.7444 |
>
> We observe only **marginal degradation** even for very small regions, indicating that our selection remains robust.
>
> **(2) On generalization beyond current benchmarks**
>
> We evaluate SIVS on **LLaVA-Bench**, which involves substantially longer outputs:
>
> |         | Score | Retention | Compression |
> | ------- | ----- | --------- | ----------- |
> | Vanilla | 45.0  | 100%      | 0%          |
> | SIVS    | 44.5  | 98.89%    | 89.78%      |
>
> Output lengths: avg 138 tokens (vs. 1–5 in typical VQA), max 881 tokens.
> These findings demonstrate the strong scalability of SIVS, showing that performance remains highly stable even as generation length increases significantly.
>
> In addition, **MileBench** in Table 1 already includes **temporally structured long-context reasoning**:
>
> | Task Category                     | LLaVA-7B | SIVS  |
> | --------------------------------- | -------- | ----- |
> | T-1 Action Understanding          | 39.33    | 39.33 |
> | T-2 Object/Scene Understanding    | 46.00    | 46.12 |
> | T-3 Navigation & Localization     | 32.25    | 31.75 |
> | T-4 Counterfactual & State Change | 38.88    | 41.88 |
>
> SIVS achieves **comparable or improved performance**, including **+3.0 on T-4**, suggesting it preserves temporally relevant information.
>
> We agree that full video benchmarks are valuable and will include them in future work.
>
> **(3) On baseline comparability**
>
> All compared methods are training-free and applied at inference time, making them directly comparable. We will clarify this in the paper.
>
> **(4) On reliability of statistics**
>
> SIVS computes all quantities online per decoding step, without dataset-level statistics. The only statistics are the mean and std of token-level semantic contributions used in the adaptive threshold mentioned in Sec. 3.3.
>
> We control token count in our setup:
> - LLaVA: fixed **576 visual tokens**
> - Qwen2.5-VL: minimum resolution → **~576 tokens**
>
> Under this setting, contribution distributions are **highly stable**, leading to consistent behavior. Moreover, the mean+std normalization is scale-invariant, adapting to relative distributions rather than absolute values.
>
> **Summary**
>
> Overall, the above results demonstrate that the concerns are effectively addressed. SIVS exhibits robust behavior under fine-grained visual conditions, generalizes to long-form and temporal reasoning, and maintains stable, fully training-free operation.

---

> > ### Author Rebuttal · Reviewer_pqa3 · 2026-04-04
> >
> > I appreciate the effort from the authors in rebuttal. My concerns are covered well. However, I remain with the current score (weak accept) as the questions and weaknesses I addressed were initially not a major concern to affect my feedback on this work, and the soundness of this paper was considered in my first score.

---

### Official Review · Reviewer_Xyo6 · 2026-03-23

**Soundness:** 2
**Presentation:** 3
**Significance:** 3
**Originality:** 3
**Overall Recommendation:** 4
**Confidence:** 3

**Summary:**

The authors propose SIVS, a training-free framework for efficient VLM inference. By projecting visual-induced hidden-state variations onto a "Semantic Lens" derived from Top-K predicted tokens, SIVS accurately selects semantically critical visual tokens. It achieves 87% KV compression and a 23.5% speedup while maintaining over 99% of original performance

**Compliance With Llm Reviewing Policy:**

Affirmed.

**Final Justification:**

The authors conducted new experiments on 72B models. The scale-agnostic results and the 0.6L depth rule effectively address my concerns regarding scalability and layer sensitivity. I am satisfied with these clarifications and have upgraded my recommendation to Weak Accept.

**Key Questions For Authors:**

Please see the weakness part.

**Strengths And Weaknesses:**

**Strengths**

- Decision-Aligned Selection: Unlike traditional attention-based heuristics, SIVS introduces a "Semantic Lens" to quantify visual necessity. By projecting hidden-state variations onto a prediction-conditioned subspace, it directly measures the impact on the final output distribution rather than relying on local activation signals.
- Superior Efficiency-Performance Trade-off: The empirical results are exceptional. SIVS achieves 87.66% visual KV compression while retaining 99.93% of the vanilla model's performance, significantly outperforming existing baselines in both latency and accuracy.

**Weaknesses**

- Limited Model Scale: The study primarily evaluates smaller architectures (3B, 7B and 13B). It remains unclear how these semantic subspaces behave in much larger models (e.g., 70B+), where representation depth and semantic abstraction may differ.
- Parameter Sensitivity: The optimal parameters are model-dependent. For instance, the best-performing intermediate layer ($l^*$) shifts from 16 in the 7B model to 20 in the 13B version.

---

> ### Author Rebuttal · Authors · 2026-03-31
>
> We thank the reviewer for the positive assessment of our decision-aligned formulation and strong efficiency–performance trade-off. We address the two main concerns below.
>
> **(1) Scalability to Larger Models**
>
> Our method is inherently **scale-agnostic**, as it operates on a **prediction-conditioned semantic subspace** rather than model-specific representations. The Semantic Lens is constructed from Top-K token embeddings, forming a **fixed low-dimensional subspace** independent of model size. Thus, SIVS depends on **local decision geometry**, not global model capacity .
>
> To directly validate scalability, we evaluate on larger-scale Qwen2.5-VL models, as LLaVA backbones are limited to 7B and 13B and thus insufficient for studying scaling behavior.
>
> **GQA**
>
> | Model | Vanilla | SIVS  | Retention | Compression |
> | ----- | ------- | ----- | --------- | ----------- |
> | 3B    | 58.89   | 58.86 | 99.95%    | 91.08%      |
> | 7B    | 60.77   | 60.76 | 99.98%    | 90.87%      |
> | 32B   | 62.06   | 61.76 | 99.52%    | 88.32%      |
> | 72B   | 62.16   | 62.26 | 100.16%   | 87.94%      |
>
> **TextVQA**
>
> | Model | Vanilla | SIVS  | Retention | Compression |
> | ----- | ------- | ----- | --------- | ----------- |
> | 3B    | 78.17   | 77.32 | 98.91%    | 88.07%      |
> | 7B    | 84.50   | 81.92 | 96.59%    | 90.80%      |
> | 32B   | 79.74   | 77.70 | 97.44%    | 88.94%      |
> | 72B   | 84.7    | 81.54 | 96.27%    | 86.05%      |
>
> Across scales, SIVS maintains **~86–91% compression** with **near-lossless performance**. No degradation is observed with increasing model size.
>
> **(2) Sensitivity to Layer Selection**
>
> SIVS does **not rely on precise layer tuning**, but on a **stable depth region**.
>
> Across all tested models, the optimal layer consistently appears near **0.6L of total depth**, specifically within **[0.56L, 0.63L]** (Qwen: **[0.61L, 0.63L]**):
>
> | Model     | Total Layers | Best ℓ* | ratio |
> | --------- | ------------ | ------- | ----- |
> | Qwen-3B   | 36           | 22      | 0.61  |
> | Qwen-7B   | 28           | 17      | 0.61  |
> | Qwen-32B  | 64           | 39      | 0.61  |
> | Qwen-72B  | 80           | 50      | 0.63  |
> | LLaVA-7B  | 32           | 18      | 0.56  |
> | LLaVA-13B | 40           | 24      | 0.60  |
>
> Importantly, performance is **not sharply peaked**: a broad mid-to-late layer range yields comparable results (Fig.2, Appendix C.1 ). This indicates SIVS depends on **the stage of representation where semantics are well formed**, rather than the exact layer index.
>
> This aligns with our analysis: early layers lack semantic abstraction, while deeper layers diminish visual influence; mid-to-late layers provide the optimal balance.
>
> Thus, the variation (e.g., ℓ*=16 vs. 20) reflects **architecture-dependent fusion depth**, not instability.
>
> **Final Remark**
>
> With the added **72B-scale experiments** and refined **layer-scaling analysis**, we demonstrate that both concerns are effectively addressed. SIVS shows consistent behavior across model scales and requires only coarse layer selection, highlighting its robustness and practicality.

---

> > ### Author Rebuttal · Reviewer_Xyo6 · 2026-04-03
> >
> > Thank you for the detailed rebuttal and new experiments on 72B models. The scale-agnostic results and the 0.6L depth rule effectively address my concerns regarding scalability and layer sensitivity. I am satisfied with these clarifications and have upgraded my recommendation to Weak Accept.

---

### Decision · Program_Chairs · 2026-04-30

**Decision:**

Accept (regular)

**Comment:**

The final reviewer scores are 4/4/4/4. Based on a careful reading of the rebuttal discussion, the reviewers expressed an overall positive attitude toward the paper, and their main concerns were addressed by the authors’ response. Since all reviewers were ultimately supportive of the submission after the rebuttal, my final recommendation is Weak Accept.